# Fibrosing Progressive Interstitial Lung Disease in Rheumatoid Arthritis: A Multicentre Italian Study

**DOI:** 10.3390/jcm12227041

**Published:** 2023-11-11

**Authors:** Marco Sebastiani, Vincenzo Venerito, Elenia Laurino, Stefano Gentileschi, Fabiola Atzeni, Claudia Canofari, Dario Andrisani, Giulia Cassone, Marlea Lavista, Francesco D’Alessandro, Caterina Vacchi, Arnaldo Scardapane, Bruno Frediani, Massimiliano Cazzato, Carlo Salvarani, Florenzo Iannone, Andreina Manfredi

**Affiliations:** 1Rheumatology Unit, Azienda Policlinico di Modena, University of Modena and Reggio Emilia, 41121 Modena, Italy; giulia.cassone@unimore.it (G.C.); caterina.vacchi@unimore.it (C.V.); andreina.manfredi@gmail.com (A.M.); 2Rheumatology Unit, Department of Precision and Regenerative Medicine-Ionian Area, University of Bari “Aldo Moro”, 70121 Bari, Italy; vincenzo.venerito@gmail.com (V.V.); marlea.lavista@uniba.it (M.L.); florenzo.iannone@uniba.it (F.I.); 3Rheumatology Unit, Department of Clinical and Experimental Medicine, University of Pisa, 56126 Pisa, Italyfrancesco.dalessandro@gmail.com (F.D.); m.cazzato@ao-pisa.toscana.it (M.C.); 4Rheumatology Unit, Azienda Ospedaliero-Universitaria Senese, Università Degli Studi di Siena, 53100 Siena, Italy; stefano.gentileschi@unisi.it (S.G.); bruno.frediani@unisi.it (B.F.); 5Rheumatology Unit, University of Messina, 98122 Messina, Italy; atzenifabiola@gmail.com; 6Rheumatology Unit, Azienda Ospedaliera San Camillo Forlanini, 00152 Roma, Italy; claudia.canofari@gmail.com; 7Respiratory Disease Unit, Azienda Policlinico di Modena, University of Modena and Reggio Emilia, 41121 Modena, Italy; darioandrisani@libero.it; 8Radiology Unit, University of Bari “Aldo Moro”, 70121 Bari, Italy; arnaldo.scardapane@uniba.it; 9Rheumatology Unit, AUSL Reggio Emilia-IRCCS, University of Reggio Emilia, 42123 Reggio Emilia, Italy; carlo.salvarani@unimore.it

**Keywords:** rheumatoid arthritis, interstitial lung disease (ILD), progressive fibrosing ILD, high-resolution computed tomography, nintedanib

## Abstract

Background: The INBUILD study demonstrated the efficacy of nintedanib in the treatment of progressive fibrosing interstitial lung disease different to idiopathic pulmonary fibrosis, including rheumatoid arthritis (RA)-related ILD. Nevertheless, the prevalence of RA-ILD patients that may potentially benefit from nintedanib remains unknown. Objectives and methods: The aim of the present multicentre study was to investigate the prevalence and possible associated factors of fibrosing progressive patterns in a cross-sectional cohort of RA-ILD patients. Results: One hundred and thirty-four RA-ILD patients with a diagnosis of RA-ILD, who were confirmed at high-resolution computed tomography and with a follow-up of at least 24 months, were enrolled. The patients were defined as having a progressive fibrosing ILD in case of a relative decline in forced vital capacity > 10% predicted and/or an increased extent of fibrotic changes on chest imaging in a 24-month period. Respiratory symptoms were excluded to reduce possible bias due to the retrospective interpretation of cough and dyspnea. According to radiologic features, ILD was classified as usual interstitial pneumonia (UIP) in 50.7% of patients, nonspecific interstitial pneumonia in 19.4%, and other patterns in 29.8%. Globally, a fibrosing progressive pattern was recorded in 36.6% of patients (48.5% of patients with a fibrosing pattern) with a significant association to the UIP pattern. Conclusion: We observed that more than a third of RA-ILD patients showed a fibrosing progressive pattern and might benefit from antifibrotic treatment. This study shows some limitations, such as the retrospective design. The exclusion of respiratory symptoms’ evaluation might underestimate the prevalence of progressive lung disease but increases the value of results.

## 1. Introduction

Interstitial lung disease (ILD) represents the most frequent extra-articular involvement in rheumatoid arthritis (RA) patients, and is becoming the main cause of mortality and quality of life impairment for these patients [1,2]. According to the current evidence, ILD can be detected in about 15–20% of RA [2,3], and it can occur in all stages of the disease, although a recent Danish study shows that about 50% of ILD diagnoses precede or are simultaneous to RA [2]. Although ILD might be subclinical in a large proportion of RA patients at the moment of diagnosis, about half of cases show a progression of lung involvement in a 2-year period [4].

The mortality risk for RA-ILD patients is 2 to 10 times higher than RA patients without ILD [2]; in RA-ILD patients, both a radiologic pattern of usual interstitial pneumonia (UIP) and a reduction in forced vital capacity over time are associated with a worst prognosis [3,4,5]. 

Recently, the INBUILD study demonstrated the efficacy of nintedanib in the treatment of progressive fibrosing ILD different than idiopathic pulmonary fibrosis (IPF), including RA-ILD [6]. Therefore, the identification of patients with progressive ILD is becoming crucial for the selection of patients who might benefit from antifibrotic therapy. Although a joint committee of the American Thoracic Society, European Respiratory Society, the Japanese Respiratory Society, and the Asociacion Latinoamericana de Torax (ATS/ERS/JRS/ALAT) has provided a definition of progressive fibrosing ILD [7], treatment with nintedanib remains limited to patients fulfilling the inclusion criteria of the INBUILD study. In fact, only RA patients with a fibrosing pattern of ILD and a progression of the lung disease in a 2-year period may be treated with nintedanib [6]. Since the number of RA-ILD patients suitable for antifibrotic drugs remains largely unknown, we aimed to investigate the prevalence of fibrosing progressive patterns in a cross-sectional multicentre Italian cohort of non-selected RA-ILD patients and the possible factors associated to a progressive fibrosing behavior of ILD.

## 2. Patients and Methods

All consecutive RA patients referred to six Italian Rheumatology Centres between July 2021 and June 2022 with a diagnosis of ILD from at least 2 years previous were enrolled in the study. All patients were classified as RA according to the 2010 EULAR/ACR classification criteria at the moment of inclusion in the study [8].

The study was approved by the local ethical committee and each patient gave their consent. 

High-resolution computed tomography (HRCT) scans were centrally assessed by an expert chest radiologist (AS) according to the Fleischner Society White Paper statement on the diagnosis of IPF [9]. The most recent chest high-resolution computed tomography (HRCT) and lung function tests (LFTs), carried out within 3 months from the last available follow-up, were compared to the HRCT and LFTs performed 24 months before.

For each patient enrolled in the study, demographic, clinical, and serological data were collected. HRCT and LFTs, including % of predicted forced vital capacity (FVC) and the % of predicted single-breath diffusing capacity of the lung for carbon monoxide (DLCO-SB) [10], were collected at the moment of the enrolment and retrospectively at baseline (twenty-four months before). 

An autoantibodies profile, including rheumatoid factor (RF), anticitrullinated peptide antibodies (ACPA), and antinuclear antibodies, was collected at the enrolment. Furthermore, previous and current RA treatments were recorded for each subject, namely methotrexate and biologic and targeted synthetic disease-modifying antirheumatic drugs (Table 1).

The HRCT pattern of disease was recorded as definite, probable usual interstitial pneumonia (UIP), or indeterminate for UIP. If a pattern indeterminate for UIP was noted, it was furtherly classified as nonspecific interstitial pneumonia (NSIP), fibrotic NSIP, and other patterns [11]. 

During the enrolment period, a total of 160 RA-ILD patients were visited in the centres involved in the study. Among them, 20 patients had a diagnosis of ILD less than 2 years ago, while another 6 patients were excluded for absence of HRCT and/or lung function tests in the last 2 years. In both cases, we did not have the possibility of evaluating the presence of a progressive fibrosing ILD according to the inclusion criteria of the study. 

According to the INBUILD inclusion criteria, patients were defined as having a progressive ILD in the case of a relative decline in FVC ≥ 10% predicted and/or a relative decline in FVC ≥ 5% predicted associated to an increased extent of fibrotic changes on chest imaging in a 24-month period [6]. Changes in respiratory symptoms, namely cough and dyspnea, were excluded from the inclusion criteria.

All participants gave their written informed consent and the present study was approved by the local institutional ethics committee.

Continuous variables were reported as median and interquartile range (IQR), while categorical variables were reported as absolute numbers and percentages. Categorical variables were analyzed by the chi-square test or Fisher’s exact test when appropriate and differences between the medians were determined using the Mann–Whitney test for unpaired samples. Clinical features were reported as dichotomic or ordinal parameters. A *p* value < 0.05 was considered statistically significant [12].

## 3. Results

One hundred and thirty-four RA-ILD patients were enrolled in the study (males/females 54/80, median age 73 years [66–79], median RA duration 10 years [6–19.25], and median ILD duration 53 months [31.75–101.5]). In 23.9% of cases, the diagnosis of ILD preceded or was concurrent to RA. See Table 1 for demographic, clinical, and serological data.

Rheumatoid factor was detected in 87.6% of patients, while anticitrullinated peptide antibodies (ACPA) were recorded in 76.2% of cases; anti-SSA antibodies were recorded in 14.9%. 

Eighty-four patients were currently or previously treated with methotrexate, while 94 were treated with at least one biologic or targeted synthetic disease modifying antirheumatic drugs (DMARDs), namely abatacept (31.8%), tumor necrosis factor inhibitors (31.3%), Janus kinase inhibitors (20.3%), rituximab (17.1%), and anti-interleukin 6 receptor antibodies (14.8%).

According to radiologic features, ILD was classified as probable or definite usual interstitial pneumonia (UIP) in 50.7% of patients, nonspecific interstitial pneumonia (NSIP) in 19.4%, and other patterns in 29.9%. A fibrosing pattern was also observed, other than in patients with UIP pattern, in 57.7% of patients with NSIP and 40% of patients with other patterns. Globally, a fibrosing pattern was detected in 73.9% of cases. 

The median FVC at the time of the first evaluation was 92% (79–113), while DLCO was 59.5% (50–72). 

A progression in ILD (relative FVC decline ≥ 10% or a progression in radiologic fibrotic involvement and a relative FVC decline ≥ 5%) was observed in 53 patients (39.6%). Among them, 25 patients had a radiologic progression in fibrotic alterations and a relative FVC decline ≥ 5%; 29 had a concurrent progression of HRCT manifestations and a clinically significant decline of FVC ≥ 10%; and 4 patients showed exclusively a deterioration in lung function by means of FVC ≥ 10%. Of interest, two patients, already treated with nintedanib, were classified as progressive according to their previous clinical history, but showed a stability of lung fibrosis at HRCT and an improvement in FVC during the last 2 years.

As expected, a significant difference in the relative decline in FVC and DLCO was recorded between progressive and non-progressive ILD. In particular, the median FVC decline was 12% (7–22) vs. 0% ((−2.9)–2), for progressive and non-progressive disease, respectively (*p* < 0.001), and DLCO decline was 9% ((−7.5)–4.25) vs. −1% (2–15.5) for progressive and non-progressive ILD, respectively (*p* < 0.001). 

Globally, a fibrosing progressive pattern was recorded in 36.6% of patients (49/134, 48.5% of patients with fibrosing pattern). In particular, 54.4% of patients with a UIP pattern, 30.8% of patients with NSIP, and 10% of patients with other HRCT ILD patterns showed a fibrosing progressive pattern (Table 2). 

Clinical and serological features of RA were analyzed to investigate a possible association with the fibrosing progressive pattern. At univariate analysis, the UIP pattern, DLCO, and the value of rheumatoid factor were significantly associated with the fibrosing progressive pattern. The UIP pattern and the value of rheumatoid factor remained independently associated with fibrosing progressive ILD, even after adjustment for age and sex (Table 3). 

## 4. Discussion

ILD complicates rheumatoid arthritis (RA) in more than 10% of cases, severely compromising the quality of life and survival of RA patients. In any case, the treatment of RA-ILD is still challenging and shared recommendations about its management are still lacking [13,14].

In the past years, different definitions of progressive fibrosing ILD have been provided, and recently, a joint committee of ATS/ERS/JRS/ALAT tried to revise this concept, with the aim of obtaining a world-wide accepted classification for these patients [7]. 

Many authors have described a preponderance of a fibrosing pattern such as UIP in patients with RA-ILD, but few data are yet available about the clinical history of these patients [15,16,17].

Clinical trials aiming to evaluate the efficacy of antifibrotic drugs, namely nintedanib and pirfenidone, have allowed for the first time the longitudinal investigation of patients with RA-ILD. Results of the two trials are not completely comparable, since enrolment criteria and endpoints were different in the two studies. The INBUILD trial evaluated the efficacy of nintedanib in patients with progressive fibrosis ILD different to IPF and enrolled patients with a progression of fibrotic radiologic findings, LFTs, or symptoms in a 2-year period; the primary endpoint of the study consisted in a difference in the FVC decline in a 52-week period between placebo and the treated group [6]. The TRIAL1 study included RA-ILD patients with a fibrotic ILD regardless of the behavior, and the primary outcome was a composite endpoint of a decline from baseline in FVC of 10% or more, or death. The study failed to reach the primary endpoint, but the difference in FVC decline in patients treated with pirfenidone or placebo was statistically significant [18]. In particular, patients with a UIP pattern enrolled in the TRAIL1 study showed a deterioration in FVC similar to that observed in patients with IPF, suggesting that UIP might be a progressive disease per se [19].

Nevertheless, the number of patients included in INBUILD is too small for obtaining conclusive answers about the response to the treatment and the prognosis of RA-ILD patients [6], and the recent ATS/ERS/JRC/ALAT guidelines conditionally recommended nintedanib for the treatment of progressive fibrosis disease in patients affected by RA-ILD who have failed standard management [7]. In the same guidelines, ATS/ERS/JRC/ALAT did not make any treatment recommendations for or against pirfenidone but recommended further research into the efficacy, effectiveness, and safety in non-IPF ILD patients manifesting PPF [7].

In this scenario, waiting for a validation of ATS/ERS/JRS/ALAT criteria, a treatment with nintedanib is limited to patients with fibrosing progressive ILD according to the INBUILD inclusion criteria [6]. Therefore, an estimation of these patients is important in planning therapeutic strategies for patients with rheumatic diseases, such as RA. 

Our study confirmed that RA is associated to a high proportion of fibrosing HRCT patterns of ILD, involving about 75% of RA-ILD patients. Among them, about half have a progressive pattern, in particular in patients with UIP pattern. Long-term follow-up could help in evaluating if a slower progression can also be observed in the other patients with a progressive pattern.

Recently, a retrospective Belgian study found a very high proportion of progressive fibrosing RA-ILD (48/89 patients) according to the INBUILD criteria. The retrospective design of this study probably induced an overestimation of progressive fibrosing lung disease. However, the authors also showed that the risk of death was twice that for RA-ILD compared with RA patients without lung involvement. Moreover, differently from this and other studies, the NSIP pattern was the most frequent (60.7%) followed by UIP (27.0%), and the proportion of a progressive fibrosing behavior was independent of the HRCT pattern [20]. 

In our study, only 30.8% of patients with NSIP pattern showed a progressive fibrosing ILD compared to about 55% of patients with UIP pattern. Analogous to that observed by Denis and co-workers, the proportion of progressive fibrosing ILD was rare in other patterns of ILD [20]. 

This observation confirms that in RA patients, the UIP pattern has an important prognostic value. Some authors reported an association between UIP pattern and increased mortality in RA [5], while in connective tissue diseases (CTDs), patients’ radiologic ILD pattern did not modify the survival of the patients [21]. Similarly, radiologic UIP pattern of ILD was not associated with fibrosing progressive ILD in CTDs [22,23], while UIP seems to be progressive in a large percentage of cases in RA-ILD [5,18].

Recently, the MUC5B rs35705950 variant has been associated with an increased risk of developing ILD in RA patients [24]. Despite the close association between the MUC5B rs35705950 variant and usual interstitial pneumonia, no association was detected between the MUC5B rs35705950 variant and FVC decline at 2 years [25].

Our study shows some limitations and advantages. To avoid bias induced by the partially retrospective collection of data, we decided not to collect data on dyspnea and cough. This may cause an underestimation of the proportion of progressive fibrosing ILD. The partially retrospective design of the study limited our ability in investigating for factors possibly associated to a progressive fibrosing pattern. For example, RA disease activity index, health assessment questionnaire, erythro-sedimentation rate, etc., were not recorded in our study, but they have been associated in some studies with an increased risk of developing RA-ILD [26,27]. In a recent Chinese study, multiple Cox regressions identified high RA disease activity, a high HRCT score at baseline, and diabetes mellitus as independent risk predictors for the progression of fibrosis in RA-ILD patients [28]. Moreover, we only enrolled patients referred to our centres in 2021–2022. Therefore, patients with severe disease who died within the 2 years before the enrolment period could have been excluded. For this reason, mortality and frequency of acute exacerbation have not been evaluated in our study. On the other side, patients have been enrolled in a consecutive manner in tertiary centres with expertise in ILD; therefore, we can be sufficiently sure that a rigorous collection of HRCT and lung functional data should also be assured for patients with mild or asymptomatic lung disease, allowing a realistic picture of clinical history of these patients. Recent data suggest that ILD might be an early complication of RA [2]; however, a systematic screening for ILD at diagnosis is not actually recommended in RA patients. Therefore, the diagnosis of RA-ILD can be delayed in many cases, explaining the interval of about 7 years between RA and ILD diagnosis observed in our study. In about a quarter of patients, the diagnosis of ILD pre-existed or was concurrent to RA, especially in patients with a more recent diagnosis, suggesting an increased interest in screening for ILD in the last years. 

A small number of patients without fibrosing ILD showed a progression of lung disease with a relative reduction in FVC ≥ 10%. Finally, about 60% of patients did not show a disease progression. Patients with a UIP pattern might benefit from therapy with antifibrotic drugs [6,18], but only long-term and large studies could clarify this point. Another point to be addressed is the possible effect of steroids, methotrexate, or biologic DMARDs on the progression of RA-ILD. The available data are contradictory about the possibility that these drugs might reduce or increase the progression of lung involvement in RA patients [29,30].

## 5. Conclusions

In conclusion, RA-ILD represents a heterogeneous condition, characterized by a severe prognosis [2,3]. A multidisciplinary approach, involving the expertise of a rheumatologist, radiologist, and pulmonologist, may contribute to choosing the best strategy for management of these patients [3,31]. The early identification and treatment of RA-ILD patients with fibrosing progressive disease could reduce the deterioration in lung function and improve the morbidity and mortality of these patients [3,32,33]. In particular, patients with a UIP pattern should be closely monitored for the high risk of disease progression.

## Figures and Tables

**Figure 1 jcm-12-07041-f001:**
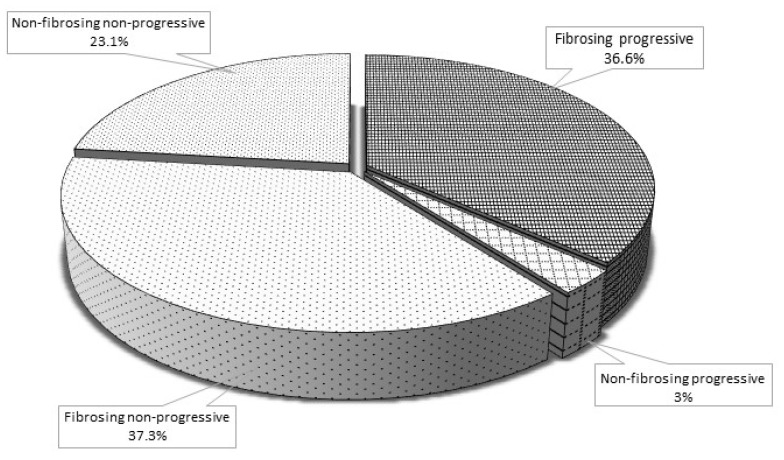
Interstitial lung disease behavior. Schematic representation of the frequencies of ILD progression patterns among the patients enrolled in the study.

**Table 1 jcm-12-07041-t001:** Clinical, demographic, and serologic features of RA-ILD patients.

	%	Number
Males/Females	40.3%/59.7%	54/80
Rheumatoid factor	84.3%	113
ACPA	71.6%	96
Anti-SSA	15.7%	21
Ever smoke	48.5%	65
	Median	IQR
Median age at last HRCT	73.0 years	66–79
Median ILD duration	53 months	31.75–101.5
Median RA duration	10 years	6–19
Age at last HRCT	72.0 years	70.4–73.7
Mean RA duration	13.1 years	11.2–15.0
Median FVC at baseline	92%	79–113
Median FVC at the end of FU	86.5%	70.75–101.25
Median DLCO at baseline	60.0%	51–72
Median DLCO at the end of FU	52.0%	44–66.5
Treatments	%	number
Ever methotrexate	66.4%	89
Ever abatacept	32%	43
Ever TNF inhibitors	31.3%	42
Ever JAK inhibitors	20.1%	27
Ever rituximab	17.2%	23
Ever anti-IL6 antibodies	14.9%	20

Continuous data are reported as median and interquartile range (IQR). ACPA: anti-cyclic citrullinated peptide antibodies; HRCT: high-resolution computed tomography; ILD: interstitial lung disease; RA: rheumatoid arthritis; FVC: forced vital capacity; DLCO: diffusion lung of carbon monoxide; FU: follow-up; TNF: tumor necrosis factor; JAK: Janus kinases; IL6: interleukin 6.

**Table 2 jcm-12-07041-t002:** Decline in FVC and DLCO according to interstitial lung disease behavior.

		Non-Fibrosing Nonprogressive	Non-Fibrosing Progressive	Fibrosing Nonprogressive	Fibrosing Progressive
Median FVC change		0 (−1.5, 3)	14 (8.5, 41)	0 (−5, 2)	12 (7, 22)
Median DLCO change		−1 (−11, 1.25)	9.1 (5.55–12.55)	−1 (−6.5, 5.25)	9 (2, 15)
HRCT patterns		UIP 0%	UIP 0%	UIP 62%	UIP 75.5%
		NSIP 22.6%	NSIP 100%	NSIP 14%	NSIP 16.3%
		Other 77.4%	Other 0%	Other 24%	Other 8.4%

FVC: forced vital capacity % predicted; DLCO: diffusion lung carbon monoxide % predicted; FU: follow-up; HRCT: high resolution computed tomography; UIP: usual interstitial pneumonia; NSIP: nonspecific interstitial pneumonia. Of interest, as above reported, a progressive behavior of disease was also recorded in 4 patients with a non-fibrosing NSIP (11.4% of patients with a non-fibrosing pattern). See Figure 1 for an overall description of the radiological and functional evolution of the enrolled patients.

**Table 3 jcm-12-07041-t003:** Association between the fibrosing progressive pattern and clinical and serological features of rheumatoid arthritis.

	Univariate Analysis			Multivariate Analysis		
	OR	95% CI	*p*	OR	95% CI	*p*
Male gender	2.01	0.98–4.12	0.056			
Rheumatoid factor	2.04	0.70–5.97	0.193			
RF value	1.002	1.000–1.003	0.035	1.002	1.000–1.005	0.030
ACPA	1.92	0.84–4.39	0.124			
ACPA value	1.000	1.000–1.001	0.278			
Anti-SSA	0.71	0.19–2.69	0.619			
Ever smoker	1.36	0.65–2.83	0.411			
Age at last HRCT	1.00	0.97–1.04	0.907			
RA duration	0.98	0.94–1.02	0.383			
ILD duration	1.00	0.99–1.01	0.250			
RA to ILD interval	1.00	0.99–1.00	0.131			
ILD preceding RA	1.73	0.72–4.16	0.218			
FVC at baseline	0.99	0.97–1.01	0.181			
DLCO at baseline	0.96	0.94–0.99	0.017	0.97	0.94–1.01	0.182
UIP pattern	5.37	2.44–11.80	<0.001	9.16	2.18–38.53	0.003
Treatments						
Ever methotrexate	0.64	0.30–1.36	0.244			
Ever abatacept	0.78	0.36–1.70	0.540			
Ever TNF inhibitors	1.59	0.74–3.39	0.239			
Ever JAK inhibitors	1.05	0.43–2.55	0.910			
Ever rituximab	1.82	0.72–4.58	0.206			
Ever anti-IL6 antibodies	0.95	0.35–2.65	0.949			

RF: rheumatoid factor; ACPA: anti-cyclic citrullinated peptides antibodies, HRCT: high-resolution computed tomography; ILD: interstitial lung disease; RA: rheumatoid arthritis; FVC: forced vital capacity; DLCO: diffusion lung of carbon monoxide; FU: follow-up; TNF: tumor necrosis factor; JAK: Janus kinases; IL6: interleukin 6.

## Data Availability

Data sharing is available if requested to the authors.

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
