# Peer review of "Fibrosing Progressive Interstitial Lung Disease in Rheumatoid Arthritis: A Multicentre Italian Study"

_jcm, 2023, doi:10.3390/jcm12227041_

Round 1

Reviewer 1 Report

Comments and Suggestions for Authors

The authors are trying to look retrospectively at a cohort of RA patients to see how often patients with RA-ILD might meet the INBUILD trial criteria and qualify for treatment with nintedanib.

Introduction could be more compelling

Design: retrospective chart review

Population: consecutive RA pts in 6 rheum centers in Italy between July 2021 and June 2022

** A chart depicting how many patients were seen in total, how many were excluded and on the basis of what, ultimately to demonstrate how they arrived at the final number of 134. This would get to the stated problem of the prevalence being unknown

Results:

**I would have appreciated a table listing the different HRCT patterns against the different parameters (PFT parameters, % extent of fibrosis on imaging, etc) at baseline and the 24 month mark. I think this may be a better way to respresent the data rather than the pie chart

Discussion:

They introduce the ATS/ERS/JRS/ALAT joint committee in the discussion, but a statement should be made in the introduction as well, since this seems to be part of what motivates the paper. Also spelling out the different acronyms is necessary.

The writers also invoke the TRAIL1 study in the discussion section just to say that the TRAIL1 population experienced a similar rate of decline in FVC. I fail to see how this is relevant to the current paper, though if the authors think it is relevant, they should first include a brief description of the TRAIL1 study (population and findings), and then articulate the relevance of that trial to this current paper better.

Conclusion could also be more compelling. The authors should articulate how their results from this descriptive paper can inform future work.

Comments on the Quality of English Language

Needs some editing

Author Response

We attached the response to reviewer 1. 

We thank the reviewer for the constructive comments that allowed to improve the manuscript

Reviewer 2 Report

Comments and Suggestions for Authors

The manuscript reports interesting data regarding fibrosing progressive ILD in RA patients.

Considerations/questions:

1. Did you consider other risk factors for development of fibrosing progressive ILD in RA patients (except smoke, ACPA) such us: ESR, RF, DAS 28, HAQ, rheumatoid nodules, radiographing changes ?

2. It would be interesting to mentioned (in discussion) the role of some MUC5B promoter variants as a risk factor for development of fibrosis in ILD-RA patients.

3. In row 110 ILD should be written instead ILS

4. Provide explanations of abbreviations below the tables

Author Response

We attached the response to reviewer 2. 

We thank the reviewer for the constructive comments that allowed to improve the manuscript
